# The representational dynamics of task and object processing in humans

**Martin N Hebart[1]\*, Brett B Bankson[1], Assaf Harel[2], Chris I Baker[1]†, Radoslaw M Cichy[3,4,5]†**

[1]Section on Learning and Plasticity, Laboratory of Brain and Cognition, National Institute of Mental Health, National Institutes of Health, Bethesda, United States; [2]Department of Psychology, Wright State University, Dayton, United States; [3]Department of Education and Psychology, Free University of Berlin, Berlin, Germany; [4]Berlin School of Mind and Brain, Humboldt Universität zu Berlin, Berlin, Germany; [5]Bernstein Center for Computational Neuroscience, Charité Universitätsmedizin, Berlin, Germany

**Abstract** Despite the importance of an observer's goals in determining how a visual object is categorized, surprisingly little is known about how humans process the task context in which objects occur and how it may interact with the processing of objects. Using magnetoencephalography (MEG), functional magnetic resonance imaging (fMRI) and multivariate techniques, we studied the spatial and temporal dynamics of task and object processing. Our results reveal a sequence of separate but overlapping task-related processes spread across frontoparietal and occipitotemporal cortex. Task exhibited late effects on object processing by selectively enhancing task-relevant object features, with limited impact on the overall pattern of object representations. Combining MEG and fMRI data, we reveal a parallel rise in task-related signals throughout the cerebral cortex, with an increasing dominance of task over object representations from early to higher visual areas. Collectively, our results reveal the complex dynamics underlying task and object representations throughout human cortex.
DOI: https://doi.org/10.7554/eLife.32816.001

**\*For correspondence:**
martin.hebart@nih.gov

†These authors contributed equally to this work

**Competing interests:** The authors declare that no competing interests exist.

## Introduction

Our tasks and behavioral goals strongly influence how we interpret and categorize the objects around us. Despite the importance of task context in our everyday perception, object recognition is commonly treated as a hierarchical feedforward process localized to occipitotemporal cortex (*Riesenhuber and Poggio, 2002*; *Serre et al., 2007*; *DiCarlo et al., 2012*) with little to no modulation by task, while categorization and task rule-related processing are mainly localized to prefrontal and parietal cortex (*Duncan, 2010*; *Freedman and Assad, 2016*). Recent fMRI work has extended this view, revealing task representations in occipitotemporal cortex, as well as documenting the impact of task on object representations in frontoparietal and occipitotemporal cortex (*Çukur et al., 2013*; *Harel et al., 2014*; *Erez and Duncan, 2015*; *Bracci et al., 2017*; *Nastase et al., 2017*; *Vaziri-Pashkam and Xu, 2017*).

While these studies demonstrate *where* in the brain task may be represented and where it may affect object representations, due to the low temporal resolution of fMRI they provide only an incomplete picture of *when* these signals emerge across different brain regions, *what* processes they reflect across the time course of a trial, and *how* task affects object representations in time. For example, are task representations first found in frontoparietal regions, first in occipitotemporal regions, or do they emerge in parallel (*Siegel et al., 2015*)? Does task affect the strength of object representations (*Peelen et al., 2009*), does it impose qualitative changes to object representations

(*Harel et al., 2014*), or both? Likewise, do task-dependent changes of object representations in occipitotemporal cortex reflect an expectation-relation top-down modulation of feedforward processing (*Kok et al., 2012*; *Kok et al., 2013*) or a late modulatory influence of task (*McKee et al., 2014*); see also *Emadi and Esteky, 2014*)?

We addressed these questions using multivariate analysis techniques on magnetoencephalography (MEG) and functional magnetic resonance imaging (fMRI) data in humans. Using multivariate decoding, we studied the temporal evolution of task and object-related brain signals and their interaction (*Carlson et al., 2013*; *van de Nieuwenhuijzen et al., 2013*; *Cichy et al., 2014*; *Isik et al., 2014*; *Clarke et al., 2015*; *Kaiser et al., 2016*). Using temporal generalization analysis (*King and Dehaene, 2014*), we probed the dynamics of the cognitive processes underlying different phases of the task. Finally, we combined MEG data with fMRI data of the same paradigm (*Harel et al., 2014*) using MEG-fMRI fusion based on representational similarity analysis (*Cichy et al., 2014*). By developing a novel model-based MEG-fMRI fusion approach, we targeted the unique contribution of task and objects to the spatiotemporal activity patterns found in human cortex.

## Results

To characterize the spatial and temporal evolution of task and object representations in the human brain, we designed a paradigm that allowed us to separately assess the effects of task and objects, as well as their interaction (*Figure 1*). Human participants (*n* = 17) categorized objects according to one of four different tasks while we monitored their brain activity using MEG. On each trial, participants were first presented with a task cue indicating the task to be carried out on an ensuing object stimulus. Two of those tasks targeted low-level perceptual dimensions (Color: red/blue, and Tilt: clockwise/counterclockwise), while the other two targeted high-level conceptual dimensions (Content: manmade/natural, Size: large/small, relative to an oven). Following the task cue, after a short delay participants were presented with an object stimulus from a set of eight different objects (five exemplars each). After another delay, a response mapping screen appeared that provided both possible answers left and right of fixation (random order). After onset of the response mapping screen, participants responded with a button press and an instructed eye blink. Participants responded fast and highly accurately (accuracy *M*: 97.19%, *SD*: 2.40; response time *M:* 712.2 ms, *SD*: 121.8), demonstrating their adaptability to the varying task demands. There were no significant behavioral differences between tasks or between objects (all *F* < 1). On average, participants missed responses or responded too slowly (RT >1,600 ms) in 1.80% of all trials (*SD*: 2.26).

### Time-resolved representation of task context and objects

All MEG analyses were carried out in a time-resolved manner. Prior to multivariate analyses, to speed up computations and increase sensitivity (*Grootswagers et al., 2017*), MEG sensor patterns (272 channels) were spatially transformed using principal component analysis, followed by removal of the components with the lowest 1% of variance, temporal smoothing (15 ms half duration at half maximum) and downsampling (120 samples/s).

To separately characterize the temporal evolution of task and object-related signals, we conducted time-resolved multivariate decoding across the trial (see *Figure 1C* and *Figure 2A*) using support vector machine classification (*Chang and Lin, 2011*) of all pairwise comparisons of conditions. For a given decoding analysis (e.g. task decoding), all pairwise classification accuracy time courses were averaged, leading to an overall chance-level of 50%. This provided temporal profiles of two resulting classification time courses, one for objects averaged across task, and one for task averaged across objects (*Figure 3—source data 1*).

In the following, we describe and report results from the 'Task Cue Period' (0 to 2,000 ms) from onset of the task cue to onset of the object stimulus, and the 'Object Stimulus Period' (2000 to 3500 ms) from onset of the object stimulus to onset of the response screen. We did not statistically analyze the ensuing 'Response-Mapping Period' (3500 ms to 5000 ms), because it was contaminated by instructed blinks and response screen-related processes (see *Materials and methods, Statistical Testing*). However, for completeness, we show results from this Response-Mapping Period in *Figures 3* and *4*.

*Task Cue Period.* As expected, classification of objects remained at chance prior to the presentation of the object stimulus (*Figure 3*, red curve). In contrast, task-related information (*Figure 3*, blue

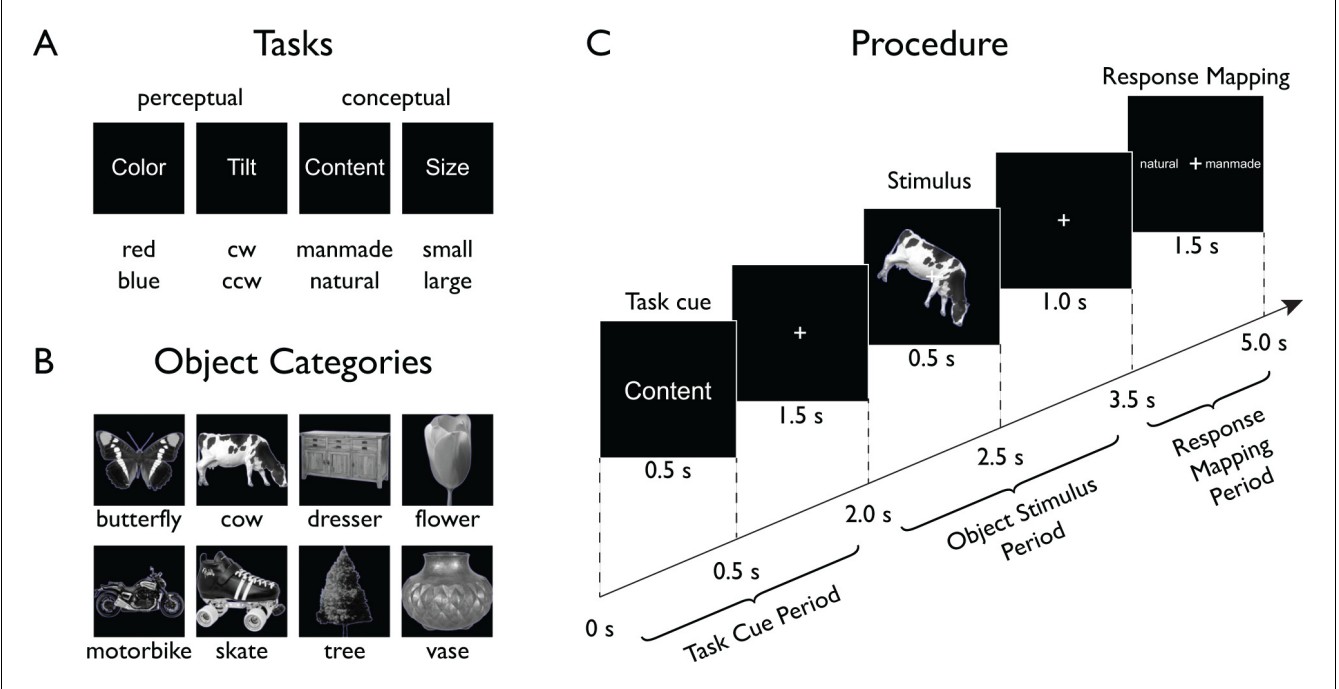

**Figure 1.** Experimental paradigm. On each trial (Procedure depicted in Panel **C**), participants were presented with a stimulus from one of eight different object classes (Panel **B**) embedded in one of four task contexts (Panel **A**, top) indicated at the beginning of each trial. Participants carried out a task that either targeted low-level features (perceptual tasks) of the object or its high-level, semantic content (conceptual tasks). After a short delay, a response-mapping screen was shown that presented the possible response alternatives (Panel **A**, bottom) in random order either left or right of fixation to decouple motor responses from the correct response.

DOI: https://doi.org/10.7554/eLife.32816.002

curve) rose rapidly in response to the task cue, peaking at 100 ms (bootstrap 95 % CI: 96–121). This was followed by a slow decay of information that approached chance-level and remained significant until ~1200 ms after cue presentation. Notably, around 1800 ms, that is, 200 ms prior to onset of the object stimulus, task information was again significantly above chance. This result demonstrates the presence of a task representation that is available prior to the onset of the object stimulus.

*Object Stimulus Period.* After onset of the object stimulus at 2000 ms, object information increased sharply, peaking after 104 ms (bootstrap 95 % CI: 100–108). This was followed by a gradual decline that remained significantly above chance until the onset of the response-mapping screen at 3500 ms. This rapid increase in object-related information was accompanied by a slow rise of task-related information starting 242 ms (bootstrap 95 % CI: 167–308) after object onset and peaking at 638 ms after object onset (95 % CI: 517–825). Information about task then remained well above-chance until the presentation of the response-mapping screen.

Together, these results demonstrate the emergence of different components of the task, including the processing of the task cue and the presence of task-related signals before and during object processing. Further, they highlight an actively maintained or reactivated task representation prior to object onset that becomes increasingly relevant during object processing. Importantly, the rise of task-related information 242 ms after object onset – more than 130 ms after peak object decoding – suggests that task context has limited impact on initial feedforward object processing, but points towards later modulation of object representations.

## Multiple stages of task processing revealed by temporal generalization analysis

The decoding of task at different time points as described above characterizes the temporal progression of task-related information across the trial. However, these results alone do not distinguish whether the decoding of task reflects a single or a sequence of multiple cognitive processes across

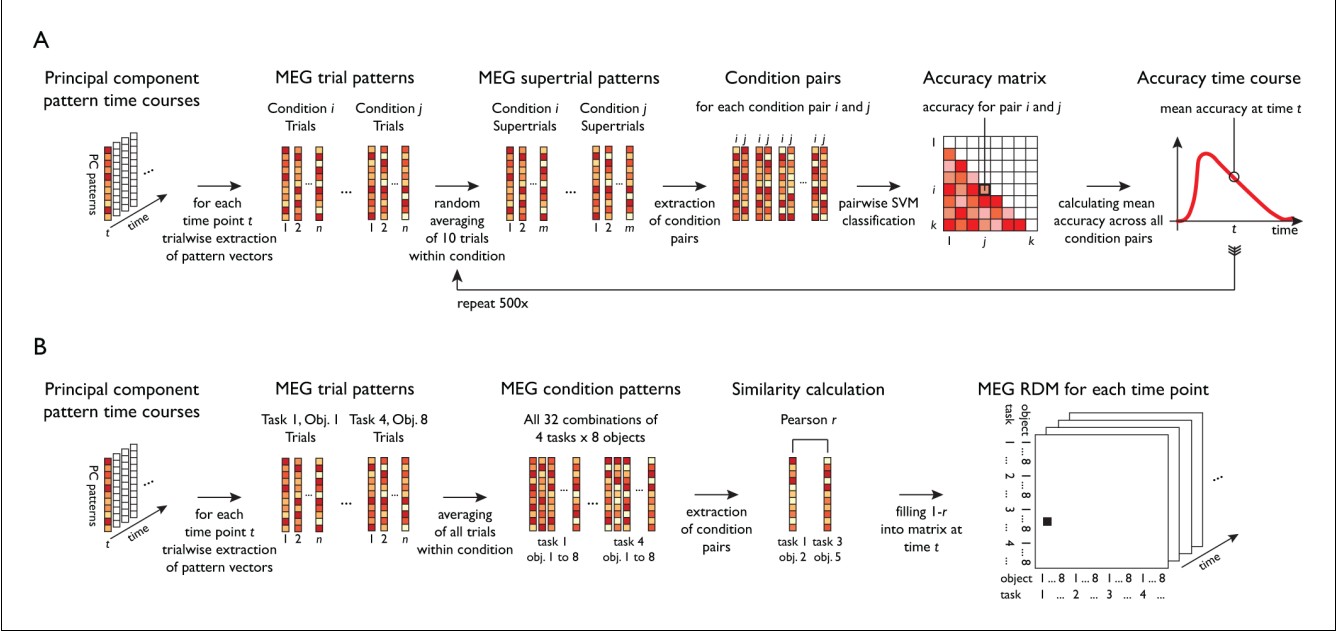

**Figure 2.** Schematic for multivariate analyses of MEG data. All multivariate analyses were carried out in a time-resolved manner on principal components (PCs) based on MEG sensor patterns (see *Materials and methods* for transformation of sensor patterns to PCs). (**A**) Time-resolved multivariate decoding was conducted using pairwise SVM classification at each time point, classifying all pairs of tasks or categories, and averaging classification accuracies within a given decoding analysis (e.g. decoding of task or category). (**B**) For model-based MEG-fMRI fusion, 32 × 32 representational dissimilarity matrices were constructed using Pearson's *r* for all combinations of task and category.

DOI: https://doi.org/10.7554/eLife.32816.003

time and more generally what cognitive processes may underlie task decoding at different time points. There are three pertinent candidates: For one, early decoding of task after task cue onset may reflect an early visual representation of the task cue that is maintained in short-term memory and accessed when the object stimulus appears in order to carry out the task. Alternatively, the task representation during object processing may reflect an abstract representation of the participant's emerging choice. Finally, the visual information about the task cue may reflect a more abstract representation of task rule that has been formed after initial visual and semantic processing of the task cue and that is maintained and applied to the object stimulus representation.

To reveal and characterize the processing stages of task, we conducted temporal generalization analysis (*Meyers et al., 2008*; *King and Dehaene, 2014*), a method to systematically analyze the similarities and differences of neural activation patterns across time. The degree to which representations are similar in different stages of the trial allows us to draw inferences about the cognitive processes involved. If a classifier can generalize from one timepoint to another, this implies similar cognitive processes at those time points. If, however, there is no temporal generalization, then this may indicate different cognitive processes. We conducted temporal generalization analysis by training a classifier at each time point during the trial to distinguish the four different tasks and then tested it at all other time points, providing us with a time ×time temporal generalization matrix.

The temporal generalization analysis revealed multiple separate, but partially overlapping stages of processing after the onset of the task cue (*Figure 4A* and *Figure 4—source data 1*, for results separated by task type see *Figure 4—figure supplement 1*, for results of a temporal generalization analysis of objects, see *Figure 4—figure supplement 2*). At a coarse level, the temporal generalization matrix exhibited a block structure within the Task Cue Period and Object Stimulus Period (*Within-Period Cross-Decoding*, *Figure 4B*, left panel). This indicates a shared representational format *within* each time period of the trial, but a largely different representational format *between* those time periods, and an abrupt change in the representational format of task after onset of the object stimulus. Importantly, this result speaks against a visual or semantic representation of task

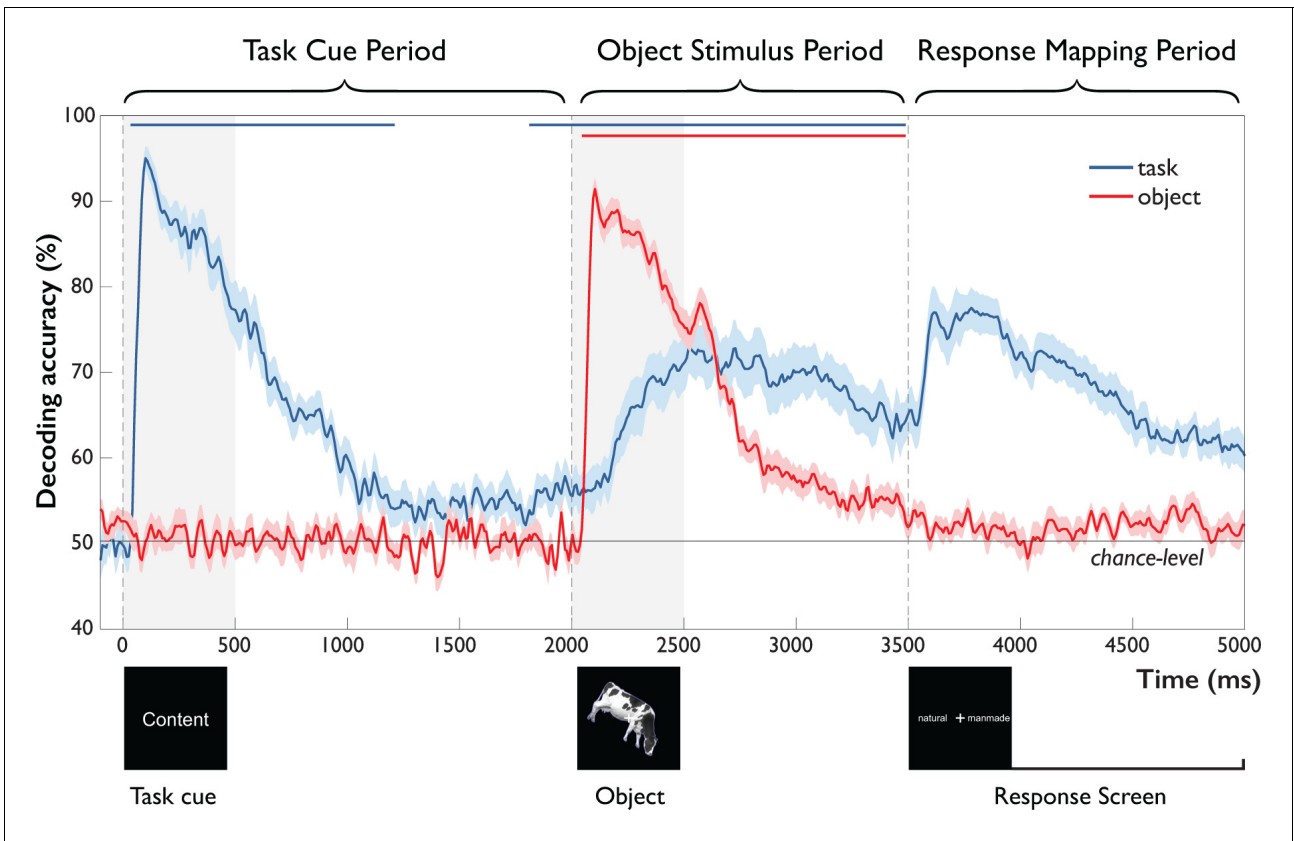

**Figure 3.** Time-resolved MEG decoding of task and objects across the trial. After onset of the task cue (Task Cue Period), task-related accuracy increased rapidly, followed by a decay toward chance and significant above-chance decoding ~200 ms prior to object onset. After onset of the object stimulus (Object Stimulus Period), object-related accuracy increased rapidly, decaying back to chance with the onset of the response-mapping screen. This was paralleled by a gradual increase in task-related accuracy, starting 242 ms and peaking 638 ms after object onset and remaining high until onset of the response-mapping screen. Error bars reflect SEM across participants for each time-point separately. Significance is indicated by colored lines above accuracy plots (non-parametric cluster-correction at p<0.05). Time periods after the onset of the response-mapping screen were excluded from statistical analyses (see *Materials and methods* and *Results*), but are shown for completeness.

DOI: https://doi.org/10.7554/eLife.32816.004

The following source data is available for figure 3:

**Source data 1.** Per subject time courses of mean classification accuracy for task and object.
DOI: https://doi.org/10.7554/eLife.32816.005

during the Object Stimulus Period, since those representations would likely have emerged already early in the Task Cue Period and would have led to between-period cross-decoding.

At a more fine-grained level, During the Task Cue Period (0 to 2000 ms) the results revealed cross-decoding lasting from ~100 ms until 2000 ms. This reinforces the idea of an active maintenance of task throughout this time period, as suggested by the time-resolved decoding analysis presented above. During the Object Stimulus Period, there was a gradual build-up of task-related information until ~200 ms after object onset. At that point, the results exhibited high levels of cross-decoding, indicating a maintained representation of task context that does not change until the onset of the response mapping screen.

Importantly, there was also evidence for a shared representational format between time periods (*Between-Period Cross-Decoding*, *Figure 4B*, middle and right panels), as demonstrated by the off-diagonals of the generalization matrix (i.e. training time 0 to 2000 ms, testing time 2000 to 3500 ms, and vice versa). First, there was generalization from the Task Cue Period to the first ~200 ms of the Object Stimulus Period (training time ~300 to 2000 ms, testing time 2000 to ~2200 ms, *Figure 4B*, middle panel), possibly reflecting a maintained short-term representation that continued until the task rule could be applied to the object. Second, there was generalization from the end of the Task

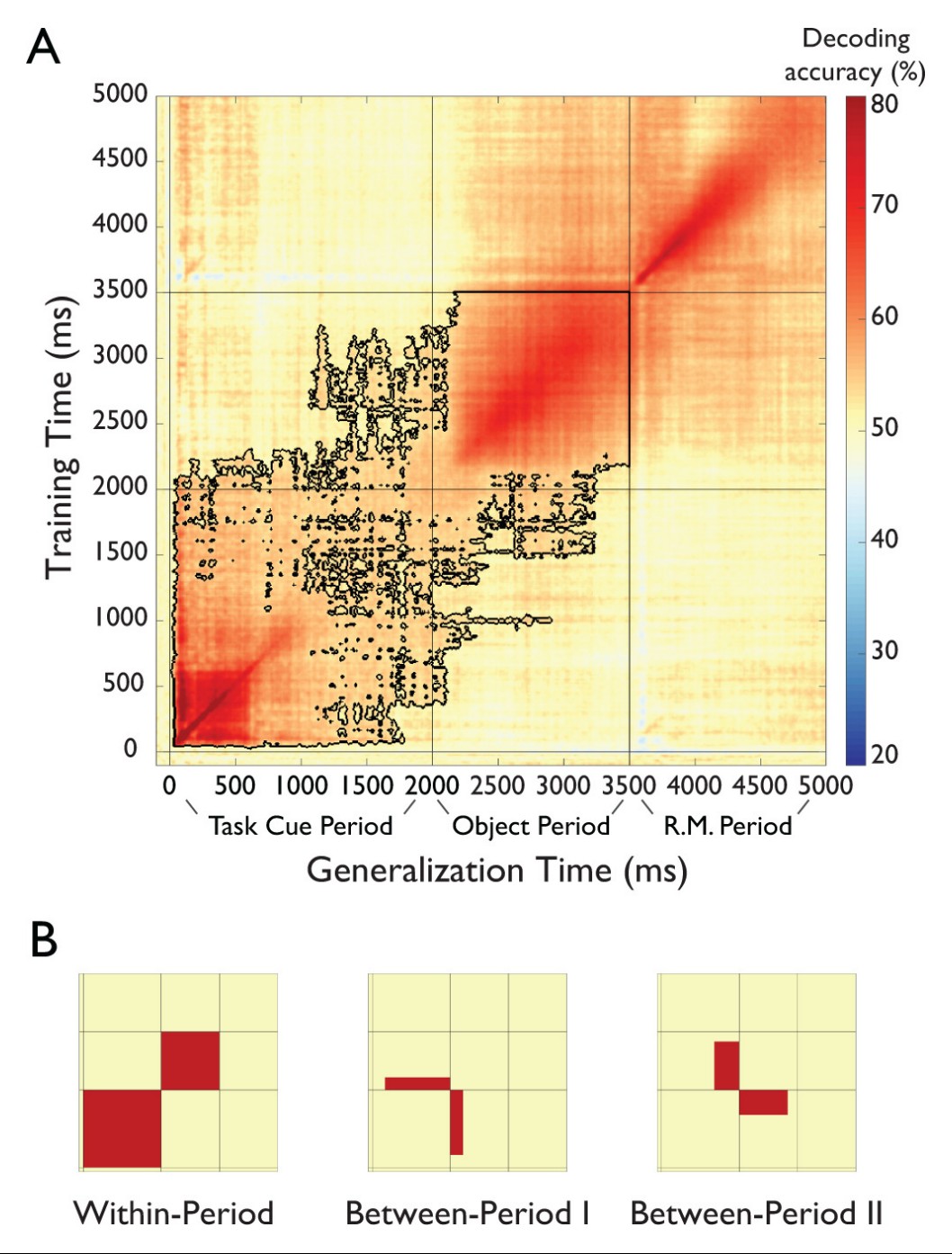

**Figure 4.** Results of temporal generalization analysis of task. (**A**) Temporal cross-classification matrix. The y-axis reflects the classifier training time relative to task cue onset, the x-axis the classifier generalization time, and the color codes the cross-classification accuracy for each combination of training and generalization time. The outline reflects significant clusters (p<0.05, cluster-corrected sign permutation test). Results after the onset of the response-mapping screen are not included in the statistical evaluation but are shown for completeness. (see *Results*) (**B**). Panels that schematically indicate three patters in the temporal generalization results. First, there was a block structure (*Within-Period Cross-Decoding*) separately spanning the Task Cue Period and the Object Stimulus Period, indicating largely different representations during the different periods of the task (left panel). At the same time, there were two separate patterns of temporal generalization in the off-diagonals (*Between-Period I and Between-Period II Cross-Decoding* illustrated in middle and right panel, respectively), indicating a shared representational format between these time periods.

*Figure 4 continued on next page*

*Figure 4 continued*

DOI: https://doi.org/10.7554/eLife.32816.006

The following source data and figure supplements are available for figure 4:

**Source data 1.** Per subject matrices of temporal cross-classification analysis of task.

DOI: https://doi.org/10.7554/eLife.32816.009

**Figure supplement 1.** Results of temporal generalization analysis of task separated by task type.

DOI: https://doi.org/10.7554/eLife.32816.007

**Figure supplement 2.** Results of temporal generalization analysis of objects.

DOI: https://doi.org/10.7554/eLife.32816.008

Cue Period to the Object Stimulus Period (training time ~1500 to 2000 ms, testing time 2000 ms to ~3300 ms, *Figure 4B*, right panel), indicating that the short-term memory representation of task was similar to the representation during application of the task rule to the object. Interestingly, this cross-classification was specific to the late short-term memory representation and did not generalize to other time points of the Task Cue Period. Note that this result cannot be explained by a representation of the correct response, because participants could not know the correct response during this short-term memory representation prior to the presentation of the object.

Together, this pattern of results suggests that the representation of task during the Object Stimulus Period likely does not reflect visual or semantic processing of the task cue (which would predict cross-classification from the early Task Cue Period); nor does it reflect only a representation of participants' choices. Rather, the results indicate that participants form an abstract representation of task rule during the short-term retention period prior to object onset, which they apply to the object stimulus when it is presented.

## Effects of task context on object representations

The robust decoding of task that increases during object processing raises the question whether the task representation is independent of object processing, to what extent task influences object representations, and when those effects emerge. Task may influence object representations in two non-exclusive ways: First, task may affect the *strength* of object representations, which would be indicated by differences in the decoding accuracy between task types. Second, task may *qualitatively* influence the representation of objects, which would be reflected in different activation patterns in response to object stimuli. These effects may emerge early (before 150 ms), indicating that task affects feedforward processing of objects. Alternatively, the effects may emerge late (after 150 ms), indicating modulatory effects of existing object representations.

To investigate whether and when task affects the strength of object representations, we conducted time-resolved multivariate decoding of objects separately for perceptual and conceptual task types and compared the time courses (*Figure 5A* and *Figure 5—source data 1*). The overall time course of object decoding was very similar for conceptual and perceptual tasks, as expected (see *Time-resolved Representation of Task Context and Objects* and *Figure 3*): decoding accuracies increased sharply after stimulus onset, followed by a gradual decline, dropping back to chance level toward the end of the Object Period. Comparing the decoding curves for conceptual and perceptual tasks directly revealed higher accuracies for conceptual tasks emerging after 542 ms (95 % CI: 283–658). In agreement with the results of the time-resolved analysis of task, this suggests that task exerts late modulatory effects on object representations, again arguing against a strong influence of task on feedforward processing. Responses to high-level conceptual tasks were more pronounced than those to low-level perceptual tasks, likely reflecting the fact that conceptual tasks entail more in-depth processing of the object than perceptual tasks.

In addition to these quantitative differences in object representations across task types, we investigated whether the object representations were qualitatively similar but differently strong (more separable patterns), or whether they were qualitatively different across task types (different patterns). To this end, we compared object classification *within* task to object classification *between* tasks. The rationale of this approach is that if the between-task cross-classification accuracy is lower than the within-task accuracy, this demonstrates that the classifier cannot rely on the same source of information in these two conditions, that is the patterns must be qualitatively different between

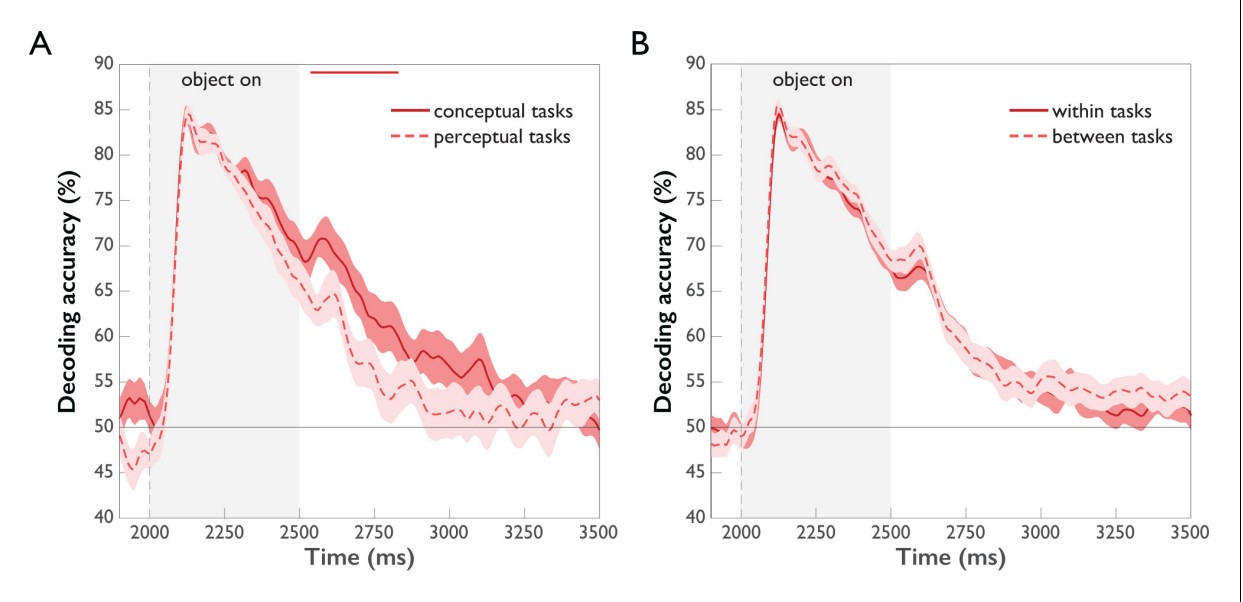

**Figure 5.** Comparison of object decoding for different task types (p<0.05, cluster-corrected sign permutation test). Error bars reflect standard error of the difference of the means. (**A**) Object decoding separated by perceptual and conceptual task types. Initially, object decoding for conceptual and perceptual tasks is the same, followed by decoding temporarily remaining at a higher level for conceptual tasks than perceptual tasks between 542 and 833 ms post-stimulus onset. (**B**) Object decoding within and across task types. A classifier was trained on data of different objects from one task type and tested either on object-related data from the same task type (within tasks) or on object-related data from the other task type (between tasks). There was no difference in within and between-task decoding.
DOI: https://doi.org/10.7554/eLife.32816.010

The following source data is available for figure 5:

**Source data 1.** Per subject time courses of mean classification accuracy for task separated by task type and cross-classification accuracy between task types.
DOI: https://doi.org/10.7554/eLife.32816.011

tasks. The results of this analysis are shown in *Figure 5B*. We found no differences in object decoding accuracies within vs. between task types, indicating that task affected only the strength, but not the quality of object representations.

## Model-based MEG-fMRI fusion for spatiotemporally-resolved neural dynamics of task and objects

Previous studies investigating task representations in humans focused primarily on the spatial localization of task effects to areas of the human brain. However, the representation of task does not emerge instantateously in all brain regions involved in processing task and is not static, but changes dynamically over time. To provide a more nuanced view of the cortical origin and the neural dynamics underlying task and object representations, we carried out MEG-fMRI fusion based on representational similarity analysis (*Figure 6A*, *Cichy et al., 2014*; *Cichy et al., 2016*). We calculated time-resolved MEG representational dissimilarity matrices (RDMs) for all combinations of task and objects (*Figure 2B*) and compared them to fMRI RDMs derived from brain activity patterns from five ROIs of a previously published study employing the same paradigm (*Harel et al., 2014*). Similarity between an fMRI RDM and MEG RDMs indicates a representational format common to that location (i.e. ROI) and those time points (for fMRI and MEG RDMs, see *Figure 6—figure supplement 1* and *Figure 6—video 1*). While previous versions of MEG-fMRI fusion reveal the shared variance between RDMs of both modalities, they leave open what portion of this variance can be attributed uniquely to specific conditions (e.g. task or objects). To overcome this limitation, we developed an approach for *model-based* MEG-fMRI fusion which not only provides a spatiotemporally resolved signal, but which also allows us to ascribe portions of this signal to the cognitive process of study. Our model-based MEG-fMRI fusion approach is based on commonality analysis (*Seibold and McPHEE, 1979*), a variance

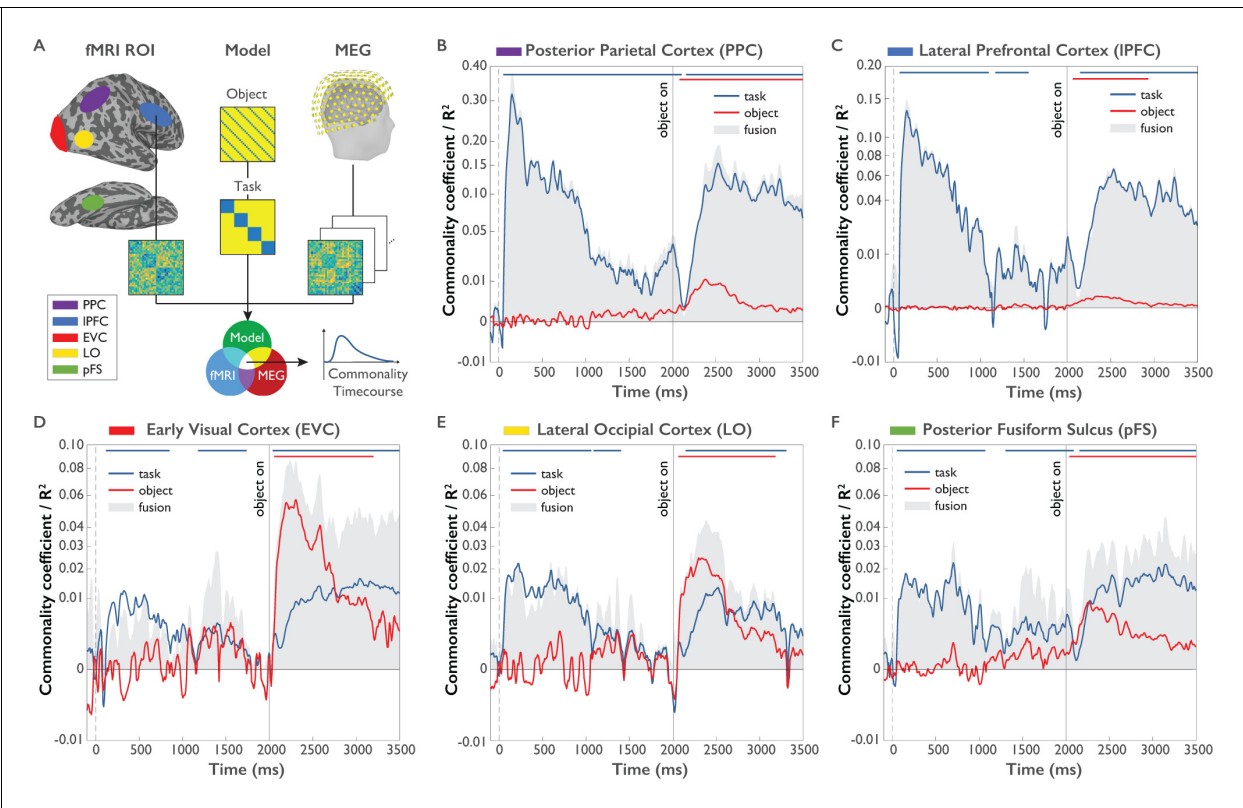

**Figure 6.** Model-based MEG-fMRI fusion procedure and results. (A) Model-based MEG-fMRI fusion in the current formulation reflects the shared variance (commonality) between three dissimilarity matrices: (1) an fMRI RDM generated from voxel patterns of a given ROI, (2) a model RDM reflecting the expected dissimilarity structure for a variable of interest (e.g. task) excluding the influence of another variable of interest (e.g. object) and (3) an MEG RDM from MEG data at a given time point. This analysis was conducted for each MEG time point independently, yielding a time course of commonality coefficients for each ROI. (B-F). Time courses of shared variance and commonality coefficients for five regions of interest (ROIs) derived from model-based MEG-fMRI fusion ($p<0.05$, cluster-corrected randomization test, corrected for multiple comparisons across ROIs): PPC (Panel B), lPFC (Panel C), EVC (Panel D), LO (Panel E) and pFS (Panel F). Blue plots reflect the variance attributed uniquely to task, while red plots reflect the variance attributed uniquely to object. Grey-shaded areas reflect the total amount of variance shared between MEG and fMRI RDMs, which additionally represents the upper boundary of the variance that can be explained by task or object models. Y-axes are on a quadratic scale for better comparability to previous MEG-RSA and MEG-fMRI fusion results reporting correlations (*Cichy et al., 2014*) and to highlight small but significant commonality coefficients.

DOI: https://doi.org/10.7554/eLife.32816.012

The following video, source data, and figure supplement are available for figure 6:

**Source data 1.** Mean representational dissimilarity matrices for all combinations of task and object, both for all five fMRI ROIs and each MEG time point, including pre-calculated permutations used for permutation testing.

DOI: https://doi.org/10.7554/eLife.32816.014

**Figure supplement 1.** FMRI representational dissimilarity matrices (RDMs) for the five regions of interest: Posterior parietal cortex (PPC), lateral prefrontal cortex (lPFC), early visual cortex (EVC), object-selective lateral occipital cortex (LO), and posterior fusiform sulcus (pFS).

DOI: https://doi.org/10.7554/eLife.32816.013

**Figure 6–video 1.** Movie of time-resolved MEG representational dissimilarity matrices, scaled using the rank transform across dissimilarities.

DOI: https://doi.org/10.7554/eLife.32816.015

partitioning approach that identifies the variance uniquely shared between multiple variables, in our case MEG, fMRI and a given model RDM (*Figure 6A*). Model RDMs were constructed based on the expected dissimilarity for task and objects (0 within the same condition, one between different conditions). The procedure results in localized time courses of task-specific and object-specific information.

The results of this model-based MEG-fMRI fusion are shown in *Figure 6B–F* separately for five regions of interest (ROIs): early visual cortex (EVC), object-selective lateral occipital cortex (LO), posterior fusiform sulcus (pFS), lateral prefrontal cortex (lPFC), and posterior parietal cortex (PPC). The

grey shaded area indicates the total amount of variance captured by MEG-fMRI fusion. Blue and red lines indicate the amount of variance in the MEG-fMRI fusion uniquely explained by the task and object model, respectively (see *Figure 6—source data 1*).

In all ROIs and at most time points, the task or object models collectively explained the majority of the shared variance between MEG and fMRI, as indicated by the close proximity of the colored lines to the upper boundary of the grey-shaded area. This result demonstrates that task and object model RDMs are well suited for describing the observed spatio-temporal neural dynamics.

All regions carried information about task and objects at some point throughout the trial, indicating that task and object representations coexist in the same brain regions, albeit not necessarily at the same point in time. Importantly, regions differed in the predominance and mixture of the represented content. Both PPC and lPFC were clearly dominated by effects of task, with much weaker object-related commonality coefficients present in these areas. These regions exhibited high-task-related commonality coefficients both during the Task Cue Period and the Object Stimulus Period. Interestingly, PPC exhibited significant task-related commonality coefficients throughout the short-term retention period that were not found in lPFC (p<0.05, cluster-corrected randomization test on differences of commonality coefficients), which may speak toward a different functional role of these regions in the retention of task rules. We found no difference between any regions in the onset of task effects after task cue onset (all p>0.05), indicating the parallel rise of task-related information in these brain regions at the temporal precision afforded by our analysis approach.

In contrast to frontoparietal regions, occipitotemporal regions EVC, LO and pFS generally exhibited weaker but significant task-related commonality coefficients than PPC and lPFC. All three regions displayed significant task-related commonality coefficients in the Task Cue Period. Interestingly, in the Object Stimulus Period all three regions exhibited a mixture of task and object-related commonality coefficients, indicating the concurrent encoding of task and objects in these brain areas. Moreover, the relative size of task-related commonalities increased gradually from EVC through LO to pFS (randomization test comparing difference of task and object representations between regions: p=0.0002), indicating an increasing importance of task encoding when progressing up the visual hierarchy. Visual inspection of the results suggests a temporal shift in the dominance of task over object representations along occipitotemporal cortex, with an earlier dominance of task in pFS than EVC. In all five regions, after onset of the object stimulus object-related commonality coefficients peaked earlier than task-related commonality coefficients (all p<0.05, based on bootstrap CI for differences in peaks), in line with the results of the time-resolved multivariate decoding analysis.

Together, we found that the spatiotemporal neural dynamics as revealed by model-based MEG-fMRI fusion predominantly reflected task or object processing, with systematic differences across cortical regions: While PPC and lPFC were dominated by task and PPC carried task information throughout the Task Cue Period, EVC, LO and pFS exhibited a mixture of task and object-related information during the Object Stimulus Period, with relative increases in the size of task-related effects when moving up the visual cortical hierarchy. This indicates the parallel representation of object and task-related signals in those brain regions, with an increasing relevance of task in category-selective brain regions.

## Discussion

We used MEG and time-resolved multivariate decoding to unravel the representational dynamics of task context, objects, and their interaction. Information about task was found rapidly after onset of the task cue and throughout the experimental trial, which was paralleled by information about objects after onset of the object stimulus. Temporal cross-decoding revealed separate and overlapping task-related processes, suggesting a cascade of representations including visual encoding of the task cue, the retention of the task rule, and its application to the object stimulus. Investigating the interaction of task context and object, we found evidence for late effects of task context on object representations, with task impacting the strength rather than the quality of object-related MEG patterns. Finally, model-based MEG-fMRI fusion revealed a parallel rise of task-related information across all regions of interest and differences in the timecourses of task and object information. Parietal and frontal regions were strongly dominated by effects of task, whereas occipitotemporal regions reflected a mixture of task and object representations following object presentation, with relative increases in task-related effects over time and along the visual cortical hierarchy.

## Representational dynamics of task context

Previous fMRI studies investigating the effects of task context during visual object processing focused on the cortical location of task effects (*Harel et al., 2014*; *Erez and Duncan, 2015*; *Bracci et al., 2017*; *Bugatus et al., 2017*; *Vaziri-Pashkam and Xu, 2017*), leaving open the question of the temporal dynamics of task representations in those regions. Our work addressed this gap in knowledge by examining the emergence of task representations and probing what cognitive processes underlie task representations at different points in time. By manipulating task context on a trial-by-trial basis we (1) mapped out the temporal evolution of task context effects across different stages of the trial, (2) uncovered different stages of processing using temporal generalization analysis, and (3) localized task-related information to different regions of the brain using model-based MEG-fMRI fusion.

The results from multivariate decoding and temporal generalization analyses indicate that following initial encoding of visual information about task cue, there was a weak but consistent short-term memory representation of this information, paralleled by an abstract representation of the task rule. Temporal generalization analysis additionally revealed multiple distinct but overlapping stages of task processing, extending previous findings of prefrontal recordings in non-human primates (*Sigala et al., 2008*; *Stokes et al., 2013*). The pattern of generalization results suggests that during object processing task is not represented in a purely visual (or semantic) format; rather, they suggest a more abstract representation of task rule that is applied to the visually-presented object stimulus (*Wallis et al., 2001*; *Stoet and Snyder, 2004*; *Bode and Haynes, 2009*; *Woolgar et al., 2011*; see also *Peters et al., 2016*).

Of note, the representation of task in monkey prefrontal cortex has been shown to be even more dynamic than described above and not to generalize at all between different periods of the task (*Stokes et al., 2013*). Since our results demonstrate phases of cross-classification between these time periods, this suggests that the source of the cross-classification between these task periods may originate from other brain regions such as posterior parietal cortex. Indeed, this interpretation is supported by our MEG-fMRI fusion results that show no significant prefrontal representations of task context during the delay period prior to the onset of the object stimulus, but a representation of task in posterior parietal cortex.

## Frontoparietal and Occipitotemporal brain areas are differentially involved in task and object representations

Previous research has suggested a dominance of parietal and prefrontal cortex in representing task context (*Duncan, 2010*; *Woolgar et al., 2011*), while the processing of objects has been attributed to occipitotemporal cortex (*Grill-Spector et al., 1999*; *Kravitz et al., 2010*; *Cichy et al., 2011*). More recently, this view has been challenged: First, object representations have been found – with some dependence on task context – in both parietal (*Konen and Kastner, 2008*; *Jeong and Xu, 2016*; *Bracci et al., 2017*; *Vaziri-Pashkam and Xu, 2017*) and prefrontal cortex (*Harel et al., 2014*; *Erez and Duncan, 2015*; *Bracci et al., 2017*). Second, there is some evidence for task effects in occipitotemporal cortex, although the extent of such effects remains debated (*Harel et al., 2014*; *Erez and Duncan, 2015*; *Lowe et al., 2016*; *Bracci et al., 2017*; *Bugatus et al., 2017*; *Vaziri-Pashkam and Xu, 2017*), and the time course of any such effects has remained elusive.

Our model-based MEG-fMRI fusion results provide a nuanced spatiotemporal characterization of task and object representations in frontoparietal and occipitotemporal cortex. Task representations emerged in parallel across all brain regions, emphasizing the importance of task representations throughout human cortex and suggesting a rapid communication of task information between brain regions (*Siegel et al., 2015*). Frontoparietal cortex was strongly dominated by task context, with much weaker object representations. This finding reinforces the notion that the dominant role of frontoparietal cortex is the representation of task, with a secondary role in representing objects (but see *Bracci et al., 2017*). In contrast, in occipitotemporal cortex, responses reflected a mixture of object and task-related effects after object onset, with an increasing dominance of task over time and along the visual cortical hierarchy from low- to high-level visual cortex (EVC, LO, pFS). These results reveal that both task and objects are encoded in parallel in the same regions of occipitotemporal cortex and suggest an increasing role of task context in high-level visual cortex.

The finding of parallel effects of task and object suggests an important role of task during object processing already in occipitotemporal cortex. This contrasts with the view of a 'passive' role of occipitotemporal cortex in the processing of objects, according to which object representations are read out by prefrontal cortex (*Freedman et al., 2003*). Instead, our results suggest that task may bias late components of object processing along occipitotemporal cortex (albeit at relatively late stages), an influence that may originate in brain regions strongly dominated by task in frontoparietal cortex (*Waskom et al., 2014*). In addition, our results suggest that this influence may increase along the visual cortical hierarchy. Indeed, pFS but not EVC or LO was found to represent task immediately prior to object onset, suggesting that task has the potential to affect the early stages of visual processing through a top-down bias. This bias may reflect a task-specific modulation of the representational strength of task-relevant object features after object onset. While this interpretation is in line with studies of attentional enhancement of objects and their features in occipitotemporal cortex (*Jehee et al., 2011*; *Peelen and Kastner, 2011*), our results go further by demonstrating the concurrent representation of both task and objects in the same brain region, which may be beneficial for optimizing the tuning of categorical brain responses to the demands of the task.

While it is possible that the MEG-fMRI fusion results found in this study are driven by a small subset of conditions or a simple one-dimensional representation, we believe this to be unlikely based on the complexity of the empirically observed MEG and fMRI RDMs (see *Figure 6—figure supplement 1* and *Figure 6—video 1*). However, future studies are required to assess the degree to which complex patterns found in multivariate analyses are driven by low-dimensional representations (*Hebart and Baker, 2017*).

## Task affects the strength of object representations late in time

The direct investigation of the temporal dynamics of task and object interactions revealed three key findings. First, we found that differences in object processing between low-level perceptual and high-level conceptual tasks emerged late in time, suggesting a late top-down modulation of object processing after initial object processing has been completed, arguing against an early expectation-related modulation of feedforward processing. This finding is consistent with a previous EEG study using natural images in an animal and vehicle detection task, reporting a fast initial object-related signal followed by later task-related responses after ~160–170 ms signaling the presence of a target stimulus (*VanRullen and Thorpe, 2001*). Similarly, a more recent MEG study (*Ritchie et al., 2015*) reported results for visual category processing in two different tasks (object categorization vs. letter discrimination) that are indicative of late differences in task-dependent stimulus processing. Finally, another recent EEG study reported late effect of task on scene processing (*Groen et al., 2016*). Overall, these combined results suggest that task representations affect late, rather than early processing of visual information.

Second, object-related information leveled off more slowly for conceptual than perceptual tasks, indicating different neural dynamics for different task types. This suggests that for conceptual tasks encoding and maintenance of object category may be beneficial for carrying out the task, in contrast to perceptual tasks for which the extraction of task-relevant features may be sufficient. Differences in the difficulty between the tasks may account for this pattern of results; however, we found no differences in response times or accuracy for the different tasks, arguing against the relevance of task difficulty. In support of this view, a previous study emploing a speeded version of the same tasks and objects found no differences in response times between tasks (*Harel et al., 2014*).

Finally, while task context affected the separability of object-related MEG patterns between task types, we found no evidence that the overall structure of those patterns changed. This result contrasts with a prior study demonstrating qualitatively different object-related patterns in lateral prefrontal and high-level object-selective cortex (*Harel et al., 2014*; *Nastase et al., 2017*). However, the contribution of multiple brain regions to the MEG response may be masking an interaction between object and task representations. Indeed, our MEG-fMRI fusion data suggest that both task and objects are being processed in parallel in pFS, although future work with independent data will be needed to resolve this issue.

While our experimental design precluded interpretation of results in the response period, future studies could explicitly target all stages of the task, from the instructional cue to the response. In addition, our study did not distinguish between different stages of object processing (e.g. low-level features or high-level categories), and our temporal generalization analysis of objects did not reveal

multiple apparent object processing stages (*Figure 4—figure supplement 2*). Task may interact with objects at any stage of processing, and while in the present study interactions arose late in time, it is still a matter of debate to what degree late responses reflect high-level categorical processing of objects (*Kaiser et al., 2016*; *Bankson et al., 2017*; *Proklova et al., 2017*). Future studies on task effects during object processing could address this issue by using a larger, controlled set of objects (*Bracci and Op de Beeck, 2016*) or by explicitly including models of shape (*Belongie et al., 2002*) or texture (*Proklova et al., 2016*). By revealing the spatiotemporal dynamics of task and object processing, our results serve as a stepping stone for future investigations addressing these questions.

## Materials and methods

### Participants

Twenty-two healthy volunteers with normal or corrected-to-normal visual acuity took part in the study. Five participants were excluded due to at least one of the following exclusion criteria: behavioral performance below 90% correct, excessive artifacts, or incomplete or corrupted recordings. Data from the remaining 17 participants (eight female, mean age 25.12, SD = 5.16) were used in all analyses throughout the study. The sample size was chosen based on previous studies employing multivariate decoding of MEG signals (*Carlson et al., 2013*; *Cichy et al., 2014*; *Cichy et al., 2016*). All participants gave written informed consent as part of the study protocol (93 M-0170, NCT00001360) prior to participation in the study. The study was approved by the Institutional Review Board of the National Institutes of Health and was conducted according to the Declaration of Helsinki.

### Experimental design and stimuli

We chose four tasks that could be carried out on a set of object images, two targeting low-level perceptual dimensions of the images, and two high-level conceptual dimensions (*Figure 1A*). The perceptual dimensions were Color (red/blue) and Tilt (clockwise/counterclockwise), and the conceptual dimensions were Content (manmade/natural) and Size (real world, large/small relative to an oven). Object images were chosen from eight different categories (*Figure 1B*): Butterfly, cow, dresser, flower, motorbike, skate, tree, and vase. For each of the eight object categories, we chose five different image exemplars. For the Color and Tilt tasks, each object was presented with a thin red or blue outline, and objects were either tilted 30 degrees clockwise or counterclockwise relative to the principal axis of the object. The combination of stimulus types led to 160 unique stimulus combinations (8 categories × 5 exemplars×2 colors×2 tilts). Each stimulus was presented once in each task context, making a total of 640 stimulus presentations per participant. The presentation order of these stimulus-task combinations was randomized. In addition, we interspersed 80 catch trials that were chosen to be random combinations of task and stimulus (see below).

All stimuli were presented on black background with a white central fixation cross present throughout the experiment. Object images were greyscale cropped images of objects and were a subset selected from a previous fMRI study (*Harel et al., 2014*). Both task cues (e.g. 'Content') and possible responses (e.g. 'manmade' or 'natural') were shown as words in white font. Task cues were always presented centrally and possible responses were shown left and right of fixation.

### Procedure

Prior to the experiment, participants were familiarized with the task by carrying out 36 randomly chosen trials outside of the MEG. For the actual experiment, participants were seated in an electro-magnetically shielded MEG chamber with their head placed in the mold of the dewar while stimuli were backprojected on a translucent screen in front of them (viewing distance: 70 cm, image size: 6 degrees of visual angle). Each trial was preceded by a white fixation cross (0.5 s) that turned green (0.5 s) to prepare participants for the upcoming trial. A trial consisted of three major components: (1) A task cue which indicated the relevant task for the trial, (2) an object stimulus which was categorized according to the task, and (3) a response-mapping screen which indicated the task-relevant response options left and right of fixation (*Figure 1C*). Based on these components, in the following we separate each trial into three different time periods: a 'Task Cue Period', an 'Object Stimulus

Period', and a 'Response Mapping Period'. Each trial lasted 5 s. A trial began with the Task Cue Period consisting of the presentation of a task cue (0.5 s) followed by a fixation cross (1.5 s). This was followed by the Object Stimulus Period consisting of the presentation of an object stimulus (0.5 s) followed by another fixation cross (1.0 s). Finally, the trial ended with the Response Mapping Period during which a response-mapping screen was displayed (1.5 s). Participants responded with the left or right index finger using an MEG-compatible response box. In addition to the button press, participants were instructed to make an eye blink during the response period to minimize the contribution of eye blink artifacts to other time periods. The order of the options on the response-mapping screen was intermixed randomly to prevent the planning of motor responses before the onset of the response screen (*Hebart et al., 2012*).

Participants were instructed to encode the task rule as soon as being presented with the task cue and to apply it immediately to the stimulus. To encourage this strategy, they were asked to respond as fast and accurately as possible. To enforce a faster application of task to object category, we introduced catch trials for which the fixation period between stimulus offset and response-mapping screen onset was shortened from 1.0 s to 0.2 s. The experiment consisted of 20 runs of 36 trials each (32 experimental trials, 4 catch trials).

## MEG recordings and preprocessing

MEG data were collected on a 275 channel CTF system (MEG International Services, Ltd., Coquitlam, BC, Canada) with a sampling rate of 1200 Hz. Recordings were available from 272 channels (dead channels: MLF25, MRF43, MRO13). Preprocessing and data analysis were carried out using Brainstorm (version 02/2016, *Tadel et al., 2011*) and MATLAB (version 2015b, The Mathworks, Natick, MA). The specifics of preprocessing and multivariate decoding (see below) were based on previously published MEG decoding work (*Cichy et al., 2014*; *Grootswagers et al., 2017*) and fine-tuned on a pilot subject that did not enter the final data set. MEG triggers were aligned to the exact presentation time on the screen that had been recorded using an optical sensor attached to the projection mirror. Data were epoched in 5.1 s trials, starting 100 ms prior to the onset of the task cue and ending with the offset of the response-mapping screen. Then, data were band-pass filtered between 0.1 and 300 Hz and bandstop filtered at 60 Hz including harmonics to remove line noise.

To further increase SNR and to reduce computational costs, we carried out (1) PCA dimensionality reduction, (2) temporal smoothing on PCA components, and (3) downsampling of the data. For PCA, data were concatenated within each channel across all trials. Note that PCA leads to orthogonal temporal components without mixing the MEG signal in time. After PCA, the components with the lowest 1% of the variance were removed, unless this would remove more than 50% of components. Since all subjects exceeded this 50% criterion, this led to 136 components for all subjects. All further analyses were conducted on the reduced set of principal components. Then, data were normalized relative to the baseline period (for task decoding: −0.1 to 0 s, for object category decoding: 1.9 to 2.0 s). To this end, for each channel we calculated the mean and standard deviation of the baseline period and subtracted this mean from the rest of the data before dividing it by the standard deviation (univariate noise normalization). Finally, the components were temporally smoothed with a Gaussian kernel of ±15 ms half duration at half maximum, and downsampled to 120 Hz (621 samples/trial).

## Time-resolved multivariate decoding

Multivariate decoding was carried out using custom-written code in MATLAB (Mathworks, Natick, MA), as well as functions from The Decoding Toolbox (*Hebart et al., 2014*), and LIBSVM (*Chang and Lin, 2011*) using linear support vector machine classification ($C = 1$). Classification was conducted for each participant separately in a time-resolved manner, that is independently for each time point. Each pattern that entered the classification procedure consisted of the principal component scores at a given time point. In the following, we describe one iteration of the multivariate classification procedure that was carried out for the example of object category classification. In the first step, we created supertrials by averaging 10 trials of the same object category without replacement (*Isik et al., 2014*). In the next step, we separated these supertrials into training and testing data, with one supertrial pattern per object category serving as test data and all other supertrial patterns as training data. This was followed by one-vs-one classification of all 28 pairwise comparisons of the

eight object categories (chance-level 50%). To test the trained classifier on the left-out data, we compared the two predicted decision values and assigned an accuracy of 100% if the order of the two test samples was predicted correctly and an accuracy of 0% if the order was the opposite (for two samples and two classes this is mathematically equivalent to the common area-under-the-curve measure of classification performance and represents a classification metric that is independent of the bias term of the classifier). In a last step, the resulting pairwise comparisons were averaged, leading to an estimate of the mean accuracy across all comparisons. This training and testing process was then repeated for each time point. This completes the description of one multivariate classification iteration for the decoding of object category. The procedure for task classification was analogous, with four tasks and six pairwise combinations. To achieve a more fine-grained and robust estimate of decoding accuracy, we ran a total of 500 such iterations of trial averaging and classification, and the final accuracy time series reflects the average across these iterations. This provided us with time-resolved estimates of MEG decoding accuracy for object category and task classification, respectively.

## Temporal generalization of task

To investigate whether the task representation remained stable across time or whether it changed, we carried out cross-classification across time, also known as temporal generalization analysis (*King and Dehaene, 2014*). The rationale of this method is that if a classifier can generalize from one time point to another, this demonstrates that the representational format is similar for these two time points. If, however, a classifier does not generalize, then under the assumption of stable noise (*Hebart and Baker, 2017*) this indicates that the representational format is different. To carry out this analysis, we repeated the same approach as described in the previous section, but instead of testing a classifier only at a given time point, we tested the same classifier for all other time points separately. This cross-classification analysis was repeated with each time point once serving as training data, yielding a time–time decoding matrix that captures classifier generalization performance across time.

## Model-based MEG-fMRI fusion for spatiotemporally-resolved information

To resolve task and category-related information both in time and space simultaneously, we carried out RSA-based MEG-fMRI fusion (*Cichy et al., 2014*; *Cichy et al., 2016*). RSA makes it possible to compare brain patterns across modalities in terms of pattern dissimilarity, abstracting from the activity patterns of measurement channels (e.g. MEG sensors) to all pairwise distances of those patterns in form of a representational dissimilarity matrices (RDMs). RSA-based MEG-fMRI fusion allows a researcher to ask the following question: At what point in time does the representational structure in a given brain area (as determined from fMRI) match the representational structure determined from the time-resolved MEG signal? The reasoning for this approach is that if the fMRI RDM of a brain region and the MEG RDM of a time point show a correspondence, this suggests that there is a shared representational format in a given brain location and at a given point in time. Here, we apply this approach to investigate the spatiotemporal evolution of object category and task representations.

FMRI RDMs for each combination of task and category ($4 \times 8 = 32 \times 32$ matrices) were available from five regions of interest (ROIs) in 25 participants who took part in a separate study employing the same task (*Harel et al., 2014*). None of these participants overlapped with the sample from the MEG study. The major difference between the MEG and the fMRI experiments were (1) that the fMRI study used an extended set of 6 tasks and (2) the exact timing of each trial was slower and jittered in the fMRI study. Details about data preprocessing have been described previously (*Harel et al., 2014*). RDMs were based on parameter estimates in a GLM for each condition which were converted to *t*-values (univariate noise normalization). Each entry in the matrix reflects one minus the correlation coefficient of the *t*-values across conditions, calculated separately for each ROI. RDMs were reduced to the relevant four task types. The five ROIs were early visual cortex (EVC), object-selective LO and pFS, lateral prefrontal cortex (lPFC) and posterior parietal cortex (PPC). EVC, LO and pFS were defined based on contrasts in an independent visual and object

localizer session, and lPFC and PPC were defined by a combination of anatomical criteria and responses in the functional localizer session to the presence of objects.

For better comparability to this previous study, we created correlation-based MEG pattern dissimilarity matrices for all combinations of task and object category. In particular, for each combination of task and category, we created a mean pattern, yielding a total 32 brain patterns per participant (8 categories × 4 tasks). We then ran a Pearson correlation between all patterns and converted these similarity estimates to dissimilarity estimates (using one minus correlation), providing us with a 32 × 32 RDM for each time point and participant.

Since different groups of participants were tested in the fMRI and MEG studies, we used the group average pattern dissimilarity matrices of each modality as the best estimate of the true pattern dissimilarity. These RDMs were symmetrical around the diagonal, so we extracted the lower triangular component of each pattern dissimilarity matrix – importantly, excluding the diagonal (*Ritchie et al., 2017*) – and converted them to vector format for further analyses, in the following referred to as representational dissimilarity vector (RDV).

For a given brain region, we conducted MEG-fMRI fusion by calculating the squared Spearman correlation between the fMRI RDV and the MEG RDV for each time point separately. The squared correlation coefficient is mathematically equivalent to the coefficient of determination ($R^2$) of the fMRI RDV explaining the MEG RDV. This approach was repeated for each fMRI RDV of the five ROIs, providing us with five time courses of representational similarity between MEG and fMRI.

While MEG-fMRI fusion provides a temporal profile of representational similarities for a given brain region, these MEG-fMRI fusion time courses do not distinguish whether MEG-fMRI representational similarities reflect task, object category, or a mixture of the two. To disentangle task and object category-related information with MEG-fMRI fusion, we extended this approach by introducing model RDMs of the same size (32 × 32). These RDMs reflected the expected dissimilarity for the representation of task and category, respectively, with entries of 1 for high expected dissimilarity (different task/category) and 0 for low expected dissimilarity (same task/category). This model-based MEG-fMRI fusion approach was carried out using commonality analysis (*Seibold and McPHEE, 1979*), a variance decomposition approach that makes it possible to estimate the shared variance between more than two variables (see *Greene et al., 2016*), for a similar approach using multiple model RDMs). For a given brain region and time point, these variables reflect (1) an MEG RDV, (2) an fMRI RDV and (3) the two model RDVs for task and object category representations.

A schematic of this model-based MEG-fMRI fusion is shown in *Figure 6A*. We conducted commonality analysis by comparing two squared semi-partial correlation coefficients (Spearman correlation), one reflecting the proportion of variance shared between MEG and fMRI partialling out all model variables excluding the variable of interest (e.g. task) from fMRI, and the other reflecting the proportion of shared variance when partialling out all model variables from fMRI including this variable of interest. The difference between both coefficients of determination ($R^2$) then provides the commonality, which is the variance shared between MEG and fMRI that is uniquely explained by the variable of interest. Formally, the commonality at time $t$ and location $j$ can be described as:

$$C_{X_t,(Y_j,A)} = R^2_{X_t,(Y_j,B)} - R^2_{X_t,(Y_j,A,B)}$$

where $X$ reflects MEG, $Y$ reflect fMRI, $A$ reflects task, and $B$ reflects object category. Note that this variable can become slightly larger than the total $R^2$ or slightly negative, due to numerical inaccuracies or the presence of small suppression effects (*Pedhazur, 1997*). In addition, commonality coefficients always reflect the shared variance relative to a target variable (in our case MEG), but depending on the relationship between the variables the estimate of shared variance can change when a different target variable is used (in our case fMRI). In the present study, the pattern of results was comparable irrespective of which variable served as a target variable.

## Statistical testing

Throughout this article, we used a non-parametric, cluster-based statistical approach to test for time periods during which the group of participants showed a significant effect (*Nichols and Holmes, 2002*), and bootstrap sampling to determine confidence intervals for peak latencies and peak latency differences. We did not compute statistics in time periods after the onset of the response-mapping screen, because (1) these time periods were corrupted by the instructed eye blinks and (2)

information about task is contained in the response-mapping screen, making it difficult to uniquely assign these responses to task or response-mapping screen. For object category-related responses we did not compute statistics for time periods prior to the onset of the object stimulus, because it was not reasonable to assume that these periods would contain information about the category before its identity is revealed. For completeness, however, we plot these results in *Figures 3* and *4*. Please note that the pattern of results reported is very similar when including these time periods into our statistical analyses.

### Non-parametric cluster-based statistical approach

We carried out a non-parametric, cluster-based statistical analysis using the maximum cluster size method (*Nichols and Holmes, 2002*). Clusters were defined as neighboring time points that all exceed a statistical cutoff (cluster-inducing threshold). This cutoff was determined through a sign-permutation test based on the distribution of *t*-values from all possible permutations of the measured accuracy values ($2^{17}$ = 131,072). The cluster-inducing threshold was defined as the 95th percentile of the distribution at each time point (equivalent to $p < 0.05$, one-sided). To identify significant clusters, we determined the 95th percentile of maximum cluster sizes across all permutations (equivalent to $p < 0.05$, one-sided). This provided us with significant clusters at the pre-specified statistical cutoffs.

For temporal generalization matrices, we extended the cluster-based approach described above to two dimensions, revealing significant 2D clusters. Because of computational limitations, we ran only a subset of 10,000 permutations drawn randomly without replacement from all available permutations.

For model-based MEG-fMRI fusion, we used an approach similar to that described above. However, instead of running a sign-permutation test across participants, we conducted a randomization test for which we created 5000 MEG similarity matrices for each of the five ROIs. These matrices were based on random permutations of the rows and columns of the group average MEG similarity matrix (*Kriegeskorte et al., 2008*). We then carried out model-based MEG-fMRI fusion using these matrices to create an estimated null distribution of information time courses for each ROI. For each time point in each ROI, a cluster-inducing threshold was determined by choosing the 95th percentile of this estimated null distribution (equivalent to $p < 0.05$, one-sided). This was followed by determining the maximum cluster sizes across all permutations as described above, but across all ROIs to correct for multiple comparisons (equivalent to $p < 0.05$, one-sided, corrected for multiple comparisons across ROIs).

### Determining confidence intervals for peak latencies

We used bootstrap sampling to estimate the 95% confidence intervals (CI) of peak latencies and peak latency differences, respectively. For each iteration of the bootstrap sampling approach, we calculated a time course based on the bootstrap sample. For multivariate decoding analyses, this was a time course of accuracy from an average of $n = 17$ time courses of decoding accuracy sampled with replacement from the pool of subjects. For MEG-fMRI fusion, this was a time course of commonality coefficients, generated by sampling $n = 17$ time courses of MEG similarity matrices from the pool of subjects with replacement, averaging them, and repeating the model-based MEG-fMRI fusion approach as described above. For each bootstrap sample time course, we then calculated timing estimates in the relevant time periods (for peak latency: timing of maximum, for peak latency difference: time difference between maxima). This process was repeated (100,000 times for multivariate decoding and 5000 times for MEG-fMRI fusion), which generated a distribution of timing estimates. The 2.5 and 97.5 percentiles of this distribution reflect the 95% confidence interval of the true timing estimate. Since we downsampled our data (bin width: 8.33 ms), the confidence intervals were conservative and overestimated by up to 16.67 ms.

## Source data and code

Data and Matlab code used for statistical analyses and producing results in the main figures is available as Source Data and Source Code (source code one file) with this article.

## Acknowledgements

We would like to thank Maryam Vaziri-Pashkam for helpful discussions and Matthias Guggenmos and Edward Silson for helpful comments on previous versions of our manuscript. This work was supported by the Intramural Research Program of the National Institutes of Health (ZIA-MH-002909) - National Institute of Mental Health Clinical Study Protocol 93 M-0170, NCT00001360, the German Research Foundation (Emmy Noether Grant CI241-1/1), and a Feodor-Lynen fellowship of the Humboldt Foundation to MNH.

## Additional information

### Funding

| Funder | Grant reference number | Author |
| --- | --- | --- |
| National Institutes of Health | ZIA-MH-002909 | Martin N Hebart<br>Brett B Bankson<br>Chris I Baker |
| Deutsche Forschungsgemeinschaft | Emmy Noether Grant CI241-1/1 | Radoslaw M Cichy |
| Alexander von Humboldt-Stiftung | Feodor-Lynen fellowship | Martin N Hebart |

The funders had no role in study design, data collection and interpretation, or the decision to submit the work for publication.

### Author contributions

Martin N Hebart, Conceptualization, Software, Formal analysis, Funding acquisition, Investigation, Visualization, Methodology, Writing—original draft, Writing—review and editing; Brett B Bankson, Investigation, Writing—review and editing; Assaf Harel, Conceptualization, Resources, Writing—review and editing; Chris I Baker, Conceptualization, Supervision, Funding acquisition, Writing—original draft, Project administration, Writing—review and editing; Radoslaw M Cichy, Conceptualization, Supervision, Funding acquisition, Methodology, Writing—original draft, Writing—review and editing

### Author ORCIDs

Martin N Hebart http://orcid.org/0000-0001-7257-428X
Brett B Bankson http://orcid.org/0000-0002-7663-3918
Chris I Baker https://orcid.org/0000-0001-6861-8964

### Ethics

Human subjects: All participants gave written informed consent as part of the study protocol (93-M-0170, NCT00001360) prior to participation in the study. The study was approved by the Institutional Review Board of the National Institutes of Health and was conducted according to the Declaration of Helsinki.

### Decision letter and Author response

Decision letter https://doi.org/10.7554/eLife.32816.021
Author response https://doi.org/10.7554/eLife.32816.022

## Additional files

### Supplementary files

• Source code 1. Matlab scripts including helper functions to produce *Figures 3–6* based on available source data.
DOI: https://doi.org/10.7554/eLife.32816.016
• Transparent reporting form

DOI: https://doi.org/10.7554/eLife.32816.017

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
