## [Decision Letter]

Thank you for submitting your article "The representational dynamics of task and object category processing in humans" for consideration by *eLife*. Your article has been favorably evaluated by David Van Essen (Senior Editor) and two reviewers, one of whom, Jody C Culham (Reviewer #1), is a member of our Board of Reviewing Editors and the other is Stefania Bracci (Reviewer #2).

The reviewers have discussed the reviews with one another and the Reviewing Editor has drafted this decision to help you prepare a revised submission.

Summary:

Both reviewers were positive about the manuscript, particularly the novel combination of MEG and fMRI to address the influence of task goals upon object recognition unfolds over time. Results complement previous work and provide novel results by characterising how task and object representations evolve and interact throughout the visual hierarchy.

Essential revisions:

1) Key methods in the main text should be clarified.

Specifically, the exact inputs into the classifiers should be explained briefly in the main text (and perhaps also more explicitly in the detailed methods). In the Cichy et al. (2014, Nat Neurosci) paper, it looks like the data put into RSMs was signal strength at each spatial channel. In the present paper, some sort of PCA was done to extract seemingly spatiotemporal components. Are the features being decoded purely spatial or is there any component of rates of temporal change? Some explanation in the main text, not just the Materials and methods, perhaps along with a schematic in the figures (as in Figures 1B and 4A of Cichy) would help readers understand key elements needed to understand this as-yet-not-that-common combination of fMRI and MEG RSA.

The commonality analysis (subsection “Model-based MEG-fMRI Fusion for Spatiotemporally-Resolved Neural Dynamics of Task and Object Category”, first paragraph) could have also been summarized more clearly.

2) Some additional discussion of the interpretation of MEG-fMRI correlations is warranted.

If the MEG features are purely spatial, then what does it mean to say that the voxelwise pattern of activation in a region measured with fMRI is correlated with a sensorwise pattern of activation in MEG. If there are univariate as well as multivariate activation changes in the region, then is this relationship somewhat trivial? As the spatial features (voxels) in fMRI that contribute to successful classification can be plotted (voxel weight maps), can the sensors in MEG that contribute to classification be plotted and compared with the fMRI maps.

3) The reviewers have some concerns about the degree to which object decoding representations reflect specific object categories (e.g., cow), or object-specific visual properties. This should be addressed in the Discussion and/or the strength of the claim for category specificity should be toned down. The authors may wish to consider an additional analysis suggested by one reviewer.

*Reviewer 1 notes:*

In Figure 1B, labels under the categories or more information in the figure legend would help (I had to look at the detailed methods to learn that the categories shown were specific categories not the general categories typically studied in ventral-stream processing (e.g., cows not animals). After seeing reviewer 2's comments, I too wonder how strong the category-specific arguments can be when the stimuli would have been even more visually similar than in most studies of category-selective processing.

*Reviewer 2 notes:*

In Figure 2, the authors show significant object decoding (regardless of task) during the object stimulus period. I wonder to what extent these object – presumably, objects within the same category share very similar visual properties (e.g., shape, texture, luminance etc.). Some of the results shown make me think that the latter interpretation might be possible. In particularly, the highest decoding accuracy is found immediately after image onset. Subsequently, decoding accuracy smoothly and constantly decreases (Figure 2). A similar pattern is also visible in Figure 4A. After object onset, during the period of highest decoding accuracy, there was no decoding difference between conceptual and perceptual tasks (this difference emerges later on), thus suggesting that this effect might be related to objects' visual attributes. To address this point the authors could run an additional temporal generalisation analysis (Figure 3A) to test pattern similarities across time for object categories. If during object onsets the results show two different clusters, results would point to different cognitive processes across time. In my view, it would be a nice addition the current analyses.

I also wonder whether the authors found similar patterns of object decoding for all 8 conditions? Based on evidence from the literature, I would expect some conditions (e.g., animals) to show better decoding than others (e.g., plants).

Related to the first comment, regarding the model-based MEG-fMRI fusion analysis shown in Figure 5, the authors state: "…EVC, LO and pFS exhibited a mixture of task and category-related information during the Object Stimulus Period, with relative increases in the size of task-related effects when moving up the visual cortical hierarchy". Wouldn't you expect the same hierarchical increasing pattern for object representations? Instead, looking at the figure, it seems that the object category model explains more variance in EVC than pFS. I wonder whether a visual model (e.g., shape – such as the geometric blur descriptor (Belongie et al., 2002), which captures object shape similarities irrespective of orientation) would do a better work – especially in EVC and LO. Maybe the authors could discuss this possibility in the Discussion section.

---

## [Author Response]

Essential revisions:1) Key methods in the main text should be clarified.Specifically, the exact inputs into the classifiers should be explained briefly in the main text (and perhaps also more explicitly in the detailed methods). In the Cichy et al. (2014, Nat Neurosci) paper, it looks like the data put into RSMs was signal strength at each spatial channel. In the present paper, some sort of PCA was done to extract seemingly spatiotemporal components. Are the features being decoded purely spatial or is there any component of rates of temporal change? Some explanation in the main text, not just the Materials and methods, perhaps along with a schematic in the figures (as in Figures 1B and 4A of Cichy) would help readers understand key elements needed to understand this as-yet-not-that-common combination of fMRI and MEG RSA.

We thank the reviewers for highlighting the need for clarification. Indeed, the methods used in this article deviated slightly from those in Cichy et al. (2014). To answer the specific question, the features being decoded are purely spatial. Following the suggestion of Grootswagers et al. (2016), we used standard PCA as a preprocessing step to reduce the dimensionality of the MEG data by removing the components with the smallest 1% of variance explained. This served the purpose of speeding up analyses and increasing the sensitivity of decoding. While PCA was indeed carried out across time, importantly this preprocessing step does not mix signals in the temporal domain, i.e. decoding and similarity analyses can still be interpreted as being temporally-resolved.

Following the reviewers’ suggestion, we now better explain the input to the classifiers in the Results section (main text) and better highlight the consequences of the preprocessing steps in the Materials and methods. In addition, we introduce a new figure (Figure 2) illustrating the methods used for decoding and generating MEG dissimilarity matrices (see Figure 6 for a schematic of model-based MEG-fMRI fusion). Finally, we extended the text in the Results section to better describe the steps of MEG-fMRI fusion (see response to next comment).

Results:

“All MEG analyses were carried out in a time-resolved manner. Prior to multivariate analyses, to speed up computations and increase sensitivity (Grootswagers et al., 2016), MEG sensor patterns (272 channels) were spatially transformed using principal component analysis, followed by removal of the components with the lowest 1% of variance, temporal smoothing (15 ms half duration at half maximum) and downsampling (120 samples / s). […] This provided temporal profiles of two resulting classification time courses, one for objects averaged across task, and one for task averaged across objects (Figure 3—source data 1).”

Materials and methods:

“For PCA, data were concatenated within each channel across all trials. Note that PCA leads to orthogonal temporal components without mixing the MEG signal in time. After PCA, the […]”

The commonality analysis (subsection “Model-based MEG-fMRI Fusion for Spatiotemporally-Resolved Neural Dynamics of Task and Object Category”, first paragraph) could have also been summarized more clearly.

We modified and expanded the text to summarize the commonality analysis more clearly in the Results section:

“Similarity between an fMRI RDM and MEG RDMs indicates a representational format common to that location (i.e. ROI) and those time points (for fMRI and MEG RDMs, Figure 6—figure supplement 1 and Figure 6—figure supplement 2). […] The procedure results in localized time courses of task-specific and object-specific information.”

2) Some additional discussion of the interpretation of MEG-fMRI correlations is warranted.If the MEG features are purely spatial, then what does it mean to say that the voxelwise pattern of activation in a region measured with fMRI is correlated with a sensorwise pattern of activation in MEG. If there are univariate as well as multivariate activation changes in the region, then is this relationship somewhat trivial? As the spatial features (voxels) in fMRI that contribute to successful classification can be plotted (voxel weight maps), can the sensors in MEG that contribute to classification be plotted and compared with the fMRI maps.

As noted above, the MEG features are purely spatial, but it is important to realize that we are correlating RDMs derived from MEG and fMRI features and not directly correlating the two signals (see Figure 6). The reviewers also raise an interesting question concerning the relationship between univariate and multivariate effects. We recently addressed the question of univariate vs. multivariate regional responses in a review/opinion article (Hebart and Baker, 2017). One conclusion in this article is that even a distributed (multivariate) representation may originate from a simple one-dimensional representation, which would be expressed as a univariate signal after appropriate (linear) spatial transformations. For that reason, we believe the question of simple or complex relationships to be a general element of representational similarity analysis, irrespective of whether responses are driven by univariate or multivariate responses. However, it is generally possible that RSA results are driven by a one-dimensional representation, which would lead to low dissimilarity for only one condition, e.g. one object alone and which could drive RSA results. To address this issue, we have added all fMRI ROI RDMs (Figure 6—figure supplement 1) and a movie of MEG RDMs (Figure 6—figure supplement 2) demonstrating the complexity of the RDMs at most times, making it unlikely that the results can be accounted for by a simple one-dimensional representation. We discuss this result and the caveats in the interpretability of results derived from multivariate analysis.

Discussion:

“While it is possible that the MEG-fMRI fusion results found in this study are driven by a small subset of conditions or a simpler one-dimensional representation, we believe this to be unlikely based on the complexity of the empirically observed MEG and fMRI RDMs (see Figure 6—figure supplement 1 and Figure 6—figure supplement 2). However, future studies are required to assess the degree to which complex patterns found in multivariate analyses are driven by low-dimensional representations (Hebart and Baker, 2017).”

In addition, to address the reviewers’ suggestion, we have made a figure of sensor pattern maps for relevant time points during task and object decoding (see Author response image 1). We used the method of Haufe et al. (2014, Neuroimage) to convert the weight maps – which are difficult to interpret (see Hebart & Baker, 2017) – to more interpretable pattern maps. While such pattern maps can give some intuition about the importance of particular sensors, their interpretability is nevertheless limited, for the following reasons. While such maps may highlight sensors implicated in successful decoding, low decoding accuracies lead to unreliable pattern map estimates. In addition, for the same reason even during time periods of high decoding accuracies sensors that pick up weak representations may not be highlighted in these maps. Further, for objects the pattern maps are the average of all 28 pairwise comparisons (6 pairwise comparisons for task) and are then averaged across participants. Any difference in the fine-scale pattern maps specific to one pair of conditions or specific to any participant would therefore wash out, even though one such map may highlight highly relevant sensors in different regions. For all those reasons, we would prefer not to directly compare those maps or draw any conclusions based on the sensor pattern maps.

3) The reviewers have some concerns about the degree to which object decoding representations reflect specific object categories (e.g., cow), or object-specific visual properties. This should be addressed in the Discussion and/or the strength of the claim for category specificity should be toned down. The authors may wish to consider an additional analysis suggested by one reviewer.

We would like to thank the reviewers for this important remark. In our revised manuscript, we address this issue in multiple ways: First, we now discuss the results more generally as object processing. Second, we address the role of low-level vs. high-level processing of objects in the Discussion section. Third, we ran the analysis suggested by reviewer 2. Please see below for a point-by-point response.

Reviewer 1 notes:In Figure 1B, labels under the categories or more information in the figure legend would help (I had to look at the detailed methods to learn that the categories shown were specific categories not the general categories typically studied in ventral-stream processing (e.g., cows not animals). After seeing reviewer 2's comments, I too wonder how strong the category-specific arguments can be when the stimuli would have been even more visually similar than in most studies of category-selective processing.

We appreciate the opportunity to improve the clarity of our manuscript and have added object labels under the objects in Figure 1. We agree with both reviewers and have now modified the text to make clear that we were not intending to make a category-specific argument. See our response to reviewer 2 below for a detailed response.

Reviewer 2 notes:In Figure 2, the authors show significant object decoding (regardless of task) during the object stimulus period. I wonder to what extent these object – presumably, objects within the same category share very similar visual properties (e.g., shape, texture, luminance etc.). Some of the results shown make me think that the latter interpretation might be possible. In particularly, the highest decoding accuracy is found immediately after image onset. Subsequently, decoding accuracy smoothly and constantly decreases (Figure 2). A similar pattern is also visible in Figure 4A. After object onset, during the period of highest decoding accuracy, there was no decoding difference between conceptual and perceptual tasks (this difference emerges later on), thus suggesting that this effect might be related to objects' visual attributes. To address this point the authors could run an additional temporal generalisation analysis (Figure 3A) to test pattern similarities across time for object categories. If during object onsets the results show two different clusters, results would point to different cognitive processes across time. In my view, it would be a nice addition the current analyses.

There are two parts to the reviewer’s comment. First, the reviewer wonders about the contribution of visual properties to the object decoding. We would like to thank the reviewer for this insightful comment. We realize that our use of the term “object category” may imply an intention to study task effects exclusively during high-level semantic / conceptual processing of objects. However, we were interested more generally in the processing of task and the processing of objects while varying task, which likely includes both low-level and high-level object processing. Based on the reviewer’s input, throughout the manuscript we have changed any mention of “*object category*” or “*category*” to “*object*”. In addition, we now discuss the question of categorical brain responses as a prospect for future studies in the Discussion section.

Second, the reviewer proposed an interesting additional analysis to investigate whether there are different phases of object processing. This analysis revealed only one large cluster of significant cross-classification (Figure 4—figure supplement 2). Note that while this result demonstrates extended time periods of temporal generalization, this does not rule out the presence of high-level categorical effects (for similar results demonstrating conceptual responses, see our recent work: Bankson, Hebart et al., 2017).

Discussion:

“In addition, our study did not distinguish between different stages of object processing (e.g. low-level features or high-level categories), and our temporal generalization analysis of objects did not reveal multiple apparent object processing stages (Figure 4—figure supplement 2). […] By revealing the spatiotemporal dynamics of task and object processing, our results serve as a stepping stone for future investigations addressing these questions.”

I also wonder whether the authors found similar patterns of object decoding for all 8 conditions? Based on evidence from the literature, I would expect some conditions (e.g., animals) to show better decoding than others (e.g., plants).

Following the reviewer’s suggestion, we have now compared object decoding for all 8 object classes, the results of which are shown in Author response image 2. Note that object decoding always depends on a comparison to other objects (e.g. accuracies for “cow” reflect the comparison to “butterfly”, “dresser” etc.), which affects the interpretability of such results. While there are slight differences between the different objects, the overall pattern of accuracy time courses is the same across objects, suggesting that task effects were comparable. While the results are interesting, we believe a more detailed analysis to be beyond the scope of the present study. We now mention this as a prospect for future studies in the Discussion section.

**Author response image 2. respfig2:** 

Related to the first comment, regarding the model-based MEG-fMRI fusion analysis shown in Figure 5, the authors state: "…EVC, LO and pFS exhibited a mixture of task and category-related information during the Object Stimulus Period, with relative increases in the size of task-related effects when moving up the visual cortical hierarchy". Wouldn't you expect the same hierarchical increasing pattern for object representations? Instead, looking at the figure, it seems that the object category model explains more variance in EVC than pFS. I wonder whether a visual model (e.g., shape – such as the geometric blur descriptor (Belongie et al., 2002), which captures object shape similarities irrespective of orientation) would do a better work – especially in EVC and LO. Maybe the authors could discuss this possibility in the Discussion section.

Indeed, the object-specific responses could have increased along the cortical hierarchy. Note, though, that the overall explained variance was relatively low in LO and pFS based on the fMRI ROI RDMs in that region, which translates to smaller *absolute* commonality coefficients in the MEG-fMRI fusion results. In other words, this result need not reflect weaker encoding (see Hebart and Baker, 2017, for interpretability of comparisons between regions), making a comparison of absolute commonality coefficients between regions non-trivial. For that reason, we prefer focusing our interpretation on *relative* comparisons within region. However, we agree with the reviewer that it is possible that object similarities were more strongly driven by low-level than high-level categorical responses. Since we removed any reference to categorical processing, we now discuss this as a prospect for future studies in the Discussion section.

Discussion:

“Task may interact with objects at any stage of processing, and while in the present study interactions arose late in time, it is still a matter of debate to what degree late responses reflect high-level categorical processing of objects (Kaiser et al., 2016; Bankson et al., 2017; Proklova et al., 2017). Future studies on task effects during object processing could address this issue by using a larger, controlled set of objects (Bracci and Op de Beeck, 2016) or by explicitly including models of shape (Belongie et al., 2002) or texture (Proklova et al., 2016).”